# Validation of a multi-ancestry polygenic risk score and age-specific risks of prostate cancer: A meta-analysis within diverse populations

Fei Chen[1†], Burcu F Darst[1,2†], Ravi K Madduri[3], Alex A Rodriguez[3], Xin Sheng[1], Christopher T Rentsch[4,5,6], Caroline Andrews[7], Wei Tang[8], Adam S Kibel[9], Anna Plym[9,10], Kelly Cho[11,12], Mohamed Jalloh[13], Serigne Magueye Gueye[13], Lamine Niang[13], Olufemi J Ogunbiyi[14], Olufemi Popoola[14], Akindele O Adebiyi[14], Oseremen I Aisuodionoe-Shadrach[15], Hafees O Ajibola[15], Mustapha A Jamda[15], Olabode P Oluwole[15], Maxwell Nwegbu[15], Ben Adusei[16], Sunny Mante[16], Afua Darkwa-Abrahams[17], James E Mensah[17], Andrew Anthony Adjei[17], Halimatou Diop[18], Joseph Lachance[19], Timothy R Rebbeck[7], Stefan Ambs[8], J Michael Gaziano[11,12,20], Amy C Justice[4,5], David V Conti[1], Christopher A Haiman[1]*

[1]Department of Population and Public Health Sciences, University of Southern California, Los Angeles, United States; [2]Public Health Sciences, Fred Hutchinson Cancer Research Center, Seattle, United States; [3]Argonne National Laboratory, Lemont, United States; [4]Yale School of Medicine, New Haven, United States; [5]VA Connecticut Healthcare System, West Haven, United States; [6]London School of Hygiene and Tropical Medicine, London, United Kingdom; [7]Harvard TH Chan School of Public Health and Division of Population Sciences, Dana Farber Cancer Institute, Boston, United States; [8]Laboratory of Human Carcinogenesis, Center for Cancer Research, National Cancer Institute, Bethesda, United States; [9]Department of Surgery, Urology Division, Brigham and Women's Hospital, Harvard Medical School, Boston, United States; [10]Department of Epidemiology, Harvard T.H. Chan School of Public Health, Boston, United States; [11]VA Boston Healthcare System, Boston, United States; [12]Division of Aging, Brigham and Women's Hospital, Boston, United States; [13]Hôpital Général Idrissa Pouye, Dakar, Senegal; [14]College of Medicine, University of Ibadan and University College Hospital, Ibadan, Nigeria; [15]College of Health Sciences, University of Abuja, University of Abuja Teaching Hospital and Cancer Science Center, Abuja, Nigeria; [16]37 Military Hospital, Accra, Ghana; [17]Korle-Bu Teaching Hospital, Accra, Ghana; [18]Laboratoires Bacteriologie et Virologie, Hôpital Aristide Le Dantec, Dakar, Senegal; [19]School of Biological Sciences, Georgia Institute of Technology, Atlanta, United States; [20]Department of Medicine, Harvard Medical School, Boston, United States

*For correspondence: haiman@usc.edu

†These authors contributed equally to this work

## Abstract

**Background:** We recently developed a multi-ancestry polygenic risk score (PRS) that effectively stratifies prostate cancer risk across populations. In this study, we validated the performance of the PRS in the multi-ancestry Million Veteran Program and additional independent studies.

**Methods:** Within each ancestry population, the association of PRS with prostate cancer risk was evaluated separately in each case–control study and then combined in a fixed-effects

inverse-variance-weighted meta-analysis. We further assessed the effect modification by age and estimated the age-specific absolute risk of prostate cancer for each ancestry population.

**Results:** The PRS was evaluated in 31,925 cases and 490,507 controls, including men from European (22,049 cases, 414,249 controls), African (8794 cases, 55,657 controls), and Hispanic (1082 cases, 20,601 controls) populations. Comparing men in the top decile (90–100% of the PRS) to the average 40–60% PRS category, the prostate cancer odds ratio (OR) was 3.8-fold in European ancestry men (95% CI = 3.62–3.96), 2.8-fold in African ancestry men (95% CI = 2.59–3.03), and 3.2-fold in Hispanic men (95% CI = 2.64–3.92). The PRS did not discriminate risk of aggressive versus nonaggressive prostate cancer. However, the OR diminished with advancing age (European ancestry men in the top decile: ≤55 years, OR = 7.11; 55–60 years, OR = 4.26; >70 years, OR = 2.79). Men in the top PRS decile reached 5% absolute prostate cancer risk ~10 years younger than men in the 40–60% PRS category.

**Conclusions:** Our findings validate the multi-ancestry PRS as an effective prostate cancer risk stratification tool across populations. A clinical study of PRS is warranted to determine whether the PRS could be used for risk-stratified screening and early detection.

**Funding:** This work was supported by the National Cancer Institute at the National Institutes of Health (grant numbers U19 CA214253 to C.A.H., U01 CA257328 to C.A.H., U19 CA148537 to C.A.H., R01 CA165862 to C.A.H., K99 CA246063 to B.F.D, and T32CA229110 to F.C), the Prostate Cancer Foundation (grants 21YOUN11 to B.F.D. and 20CHAS03 to C.A.H.), the Achievement Rewards for College Scientists Foundation Los Angeles Founder Chapter to B.F.D, and the Million Veteran Program-MVP017. This research has been conducted using the UK Biobank Resource under application number 42195. This research is based on data from the Million Veteran Program, Office of Research and Development, and the Veterans Health Administration. This publication does not represent the views of the Department of Veteran Affairs or the United States Government.

## Editor's evaluation

This article is mainly for an audience of genetic epidemiologists interested in the evaluation and portability of polygenic scores. The authors rigorously estimate the association of their multi-ancestry polygenic risk scores (PRS) for prostate cancer across multiple ancestries in a meta-analysis and show effect modification by age. The authors show that their PRS is effective in risk stratification for prostate cancer.

## Introduction

Prostate cancer is the second leading cause of cancer death and represents one of the largest health disparities in the United States, with African ancestry men having the highest incidence rates (*Howlader, 2021*). Genetic factors play an important role in prostate cancer susceptibility (*Mucci et al., 2016*; *Conti et al., 2021*) and racial/ethnic disparities in disease incidence (*Conti et al., 2021*). Polygenic risk scores (PRS), comprised of common genetic variants, have been shown to enable effective risk stratification for many common cancers (*Kachuri et al., 2020*; *Mars et al., 2020*; *Balavarca et al., 2020*; *Pal Choudhury et al., 2020*). We recently conducted a multi-ancestry genome-wide association study (GWAS), including 107,247 prostate cancer cases and 127,006 controls (75.8% of European ancestry, 11.7% of East Asian ancestry, 9.1% of African ancestry, and 3.4% Hispanic), where 269 common genetic variants were genome-wide significantly associated with prostate cancer risk (*Conti et al., 2021*). Although individual genetic variants modulate disease risk only marginally, the aggregated effect of these 269 risk variants, measured by a PRS, was found to stratify prostate cancer risk in independent samples of European and African ancestry (*Conti et al., 2021*; *Plym et al., 2022*). As a measure of genetic susceptibility to prostate cancer, the PRS could potentially be an effective tool to identify men across diverse populations at higher risk of developing prostate cancer and allow them to make more informed decisions regarding at what age(s) and how frequently to undergo prostate-specific antigen (PSA) screening.

In this investigation, we evaluated the previously developed multi-ancestry PRS in large independent samples of men from the Veteran Affairs Million Veteran Program (MVP; 21,078 cases and 284,177 controls, including 13,643 cases and 210,214 controls of European ancestry, 6353 cases and

53,362 controls of African ancestry, and 1082 cases and 20,601 controls from Hispanic populations) (*Gaziano et al., 2016*), the Men of African Descent and Carcinoma of the Prostate (MADCaP) Network (405 cases and 396 controls of African ancestry) (*Harlemon et al., 2020*), and the Maryland Prostate Cancer Case–Control Study (NCI-MD; 383 cases and 395 controls of African ancestry) (*Smith et al., 2017*; 'Methods'). We also included, through meta-analysis, independent replication studies of the multi-ancestry PRS conducted to date in European (UK Biobank and Mass General Brigham [MGB] Biobank) and African ancestry populations (California and Uganda Prostate Cancer Study [CA UG] and MGB Biobank; 'Methods'; *Conti et al., 2021*; *Plym et al., 2022*), bringing the total sample to 31,925 cases and 490,507 controls.

In each of the replication studies included in our analysis, the PRS was constructed by summing variant-specific weighted allelic dosages of the 269 prostate cancer risk variants using the multi-ancestry conditional weights generated from our previous GWAS for prostate cancer ('Methods'). Within each ancestry population, the association of PRS on prostate cancer risk was evaluated separately in each study and combined in a fixed-effects inverse-variance-weighted meta-analysis. Age-stratified analyses were performed in two large replication studies, UK Biobank and MVP, to assess the age-specific effects of PRS on prostate cancer risk. The absolute risk of prostate cancer was calculated for a given age for each PRS category in men from European, African and Hispanic populations

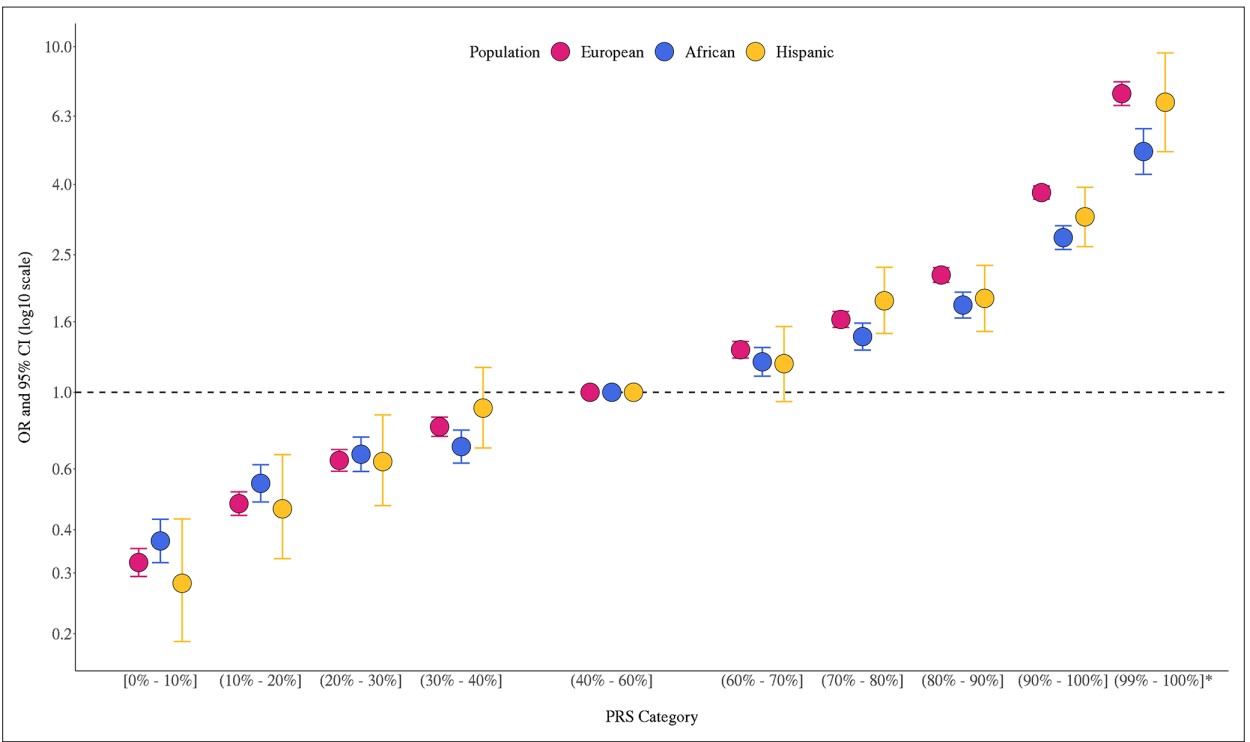

**Figure 1.** Association between the multi-ancestry polygenic risk score (PRS) of 269 variants and prostate cancer risk in men from European, African, and Hispanic populations. The European ancestry replication studies included Million Veteran Program (MVP), UK Biobank (Conti, Darst et al., *Nature Genetics*, 2021), and Mass General Brigham (MGB) Biobank (Plym et al., *JNCI*, 2021). The African ancestry replication studies included MVP, California and Uganda Prostate Cancer Study (CA UG) (Conti, Darst et al., *Nature Genetics*, 2021), Men of African Descent and Carcinoma of the Prostate (MADCaP) Network, Maryland Prostate Cancer Case–Control Study (NCI-MD), and MGB Biobank (Plym et al., *JNCI*, 2021). Replication in Hispanic men was conducted in MVP. Results from individual replication studies are shown in *Figure 1—figure supplement 1*. The x-axis indicates the PRS category. Additional analysis was performed to evaluate the PRS association in men with extremely high genetic risk (99–100%). The y-axis indicates OR with error bars representing 95% CIs for each PRS category compared to the 40–60% PRS. The dotted horizontal line corresponds to an OR of 1. ORs and 95% CIs for each decile are provided in *Figure 1—source data 1*.

The online version of this article includes the following source data and figure supplement(s) for figure 1:

**Source data 1.** Association between the multi-ancestry polygenic risk score (PRS) and prostate cancer risk replicated in men from European, African, and Hispanic populations.

**Figure supplement 1.** Association between the multi-ancestry polygenic risk score (PRS) of 269 variants and prostate cancer risk from individual replication studies of European (**A**) and African ancestry (**B**).

(*Antoniou et al., 2010*; *Kuchenbaecker et al., 2017*; *Amin Al Olama et al., 2015*; *Antoniou et al., 2001*) using age- and population-specific prostate cancer incidence from the Surveillance, Epidemiology, and End Results (SEER) Program (1999–2013) and age- and population-specific mortality rates from the National Center for Health Statistics, CDC (1999–2013). The PRS was also tested for association with disease aggressiveness in MVP ('Methods,' *Appendix 1—figure 1*).

## Results

The multi-ancestry PRS was strongly associated with prostate cancer risk in the three populations (*Figure 1*, *Figure 1—source data 1*). In European ancestry men, ORs were 3.78 (95% CI = 3.41–3.81) and 7.32 (95% CI = 6.76–7.92) for men in the top PRS decile (90–100%) and top percentile (99–100%), respectively, compared to men with average genetic risk (40–60% PRS category). In African ancestry men, ORs were 2.80 (95% CI = 2.49–2.95) and 4.98 (95% CI = 4.27–5.79) for men in the top PRS decile and percentile, respectively. In Hispanic men, ORs were 3.22 (95% CI = 2.64–3.92) and 6.91 (95% = 4.97–9.60) for men in the top PRS decile and percentile, respectively. PRS associations within each ancestry population were generally consistent across individual replication studies (*Figure 1—figure supplement 1*). The area under the curve (AUC) increased 0.136 on average across populations upon adding the PRS to a base model of age and principal components of ancestry (*Appendix 1—table 1*). Compared to the mean PRS in European ancestry controls, African ancestry controls had a mean PRS associated with a relative risk of 2.19 (95% CI = 2.17–2.21), while Hispanic controls had a relative risk of 1.16 (95% CI = 1.15–1.18), consistent with previous findings (*Conti et al., 2021*).

Previously, we found that PRS associations were significantly stronger in younger men (aged ≤ 55 years) than in older men (aged > 55 years) (*Conti et al., 2021*). In the two large replication studies, UK Biobank and MVP, we further explored effect modification by age (*Figure 2*, *Figure 2—figure supplement 1*, *Figure 2—source data 1*). In European ancestry men, for the top PRS decile, the OR was 7.11 (95% CI = 5.82–8.70) in men aged ≤55, 4.26 (95% CI = 3.77–4.81) in men aged 55–60, and 2.79 (95% CI = 2.50–3.11) in men aged >70. The gradient in PRS risk by age was greater for men in the top PRS percentile, with ORs of 17.2 (95% CI = 13.0–22.8), 9.18 (95% CI = 7.52–11.2), and 5.43 (95% CI = 4.50–6.55) estimated for men ≤55, 55–60, and >70 years of age, respectively. Attenuation of PRS associations with age was also observed in African ancestry men as the OR for men in the top PRS decile decreased from 3.75 (95% CI = 3.04–4.64) in men aged ≤55 to 2.16 (95% CI = 1.76–4.68) in men aged >70. For African ancestry men in the top PRS percentile, the OR decreased from 8.80 (95% CI = 6.16–12.6) in men aged ≤55 to 2.87 (95% CI = 1.76–4.68) in men aged >70. A similar trend was observed in Hispanic men (OR = 6.37, 95% CI = 3.26–12.44 for men ≤55 and OR = 2.15, 95% CI = 1.39–3.32 for men >70 in the top PRS decile). Compared to men in the 40–60% PRS category, men from European, African, and Hispanic populations in the top PRS decile reached 5% absolute risk of prostate cancer 12 years earlier (age 57 vs. 69), 8 years earlier (age 55 vs. 63), and 11 years earlier (age 60 vs. 71), respectively (*Table 1*, *Figure 3*). For men in the top PRS percentile, 5% absolute risk was reached by ages 51, 52, and 53 for European, African, and Hispanic populations, respectively.

Similar to previous findings (*Conti et al., 2021*; *Plym et al., 2022*), the multi-ancestry PRS did not consistently differentiate aggressive and nonaggressive prostate cancer risk (*Appendix 1—table 2*). For men in the top PRS decile, ORs were 3.17 (95% CI = 2.77–3.63) and 3.71 (95% CI = 3.48–3.94) for aggressive and nonaggressive prostate cancer in comparison to controls, respectively, in European ancestry men (P-heterogeneity = 0.04), and 1.92 (95% CI = 1.17–3.15) and 3.30 (95% CI = 2.64–4.12), respectively, in Hispanic men (P-heterogeneity = 0.05). In African ancestry men, the association was greater for aggressive (OR = 3.31, 95% CI = 2.71–4.03) than nonaggressive disease (OR = 2.66, 95% CI = 2.43–2.92), although confidence intervals overlapped (P-heterogeneity = 0.05).

## Discussion

Findings from this investigation provide further support for the PRS as a prostate cancer risk stratification tool in men from European, African, and Hispanic populations. Notably, this investigation provides the first evidence of replication of the multi-ancestry PRS in Hispanic men. Consistent with previous findings (*Conti et al., 2021*; *Plym et al., 2022*), we observed lower PRS performance in African versus European ancestry men, supporting the need to expand GWAS and fine-mapping efforts in African ancestry men. The stronger association of the PRS with prostate cancer risk observed

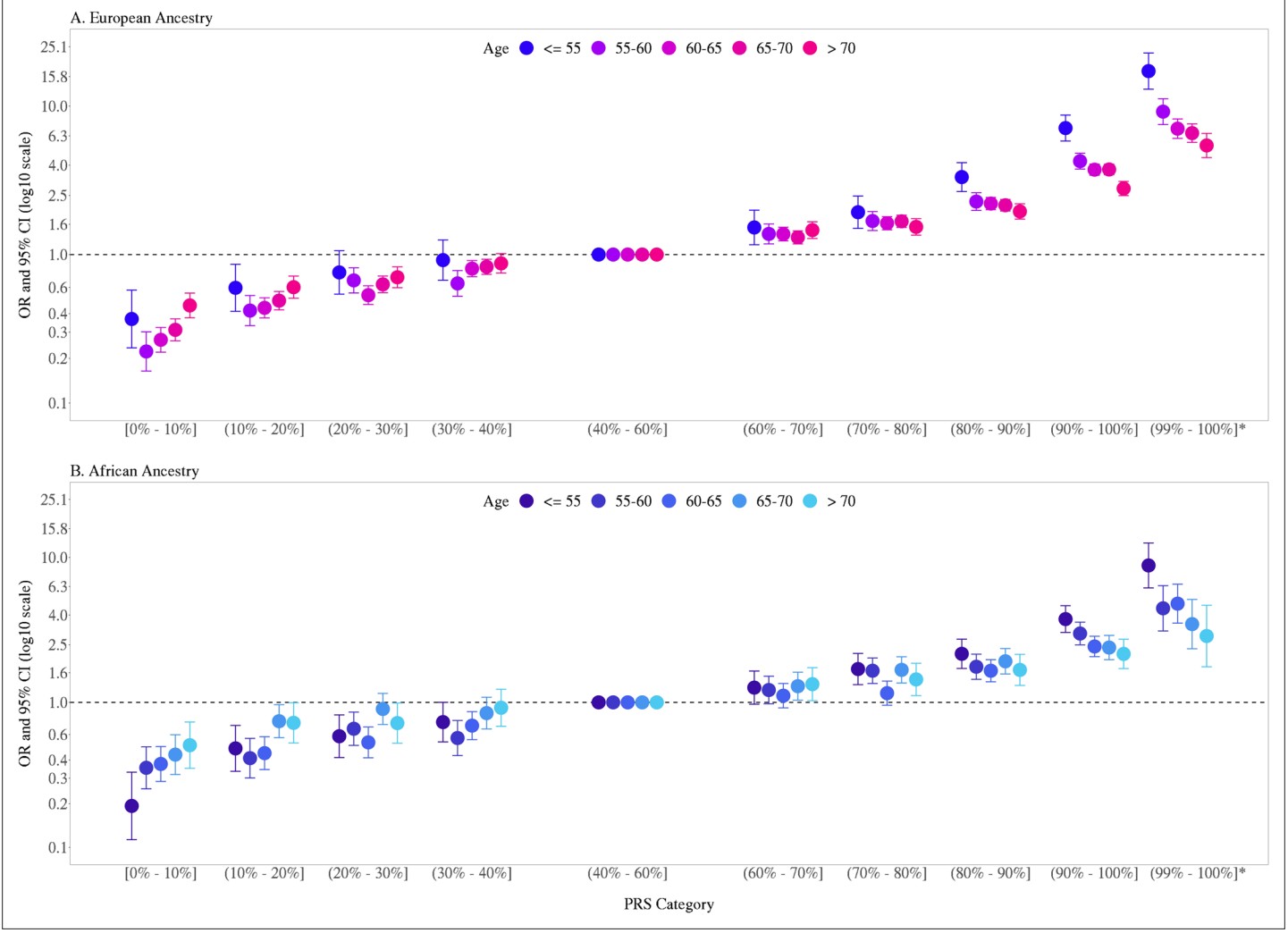

**Figure 2.** Association between the multi-ancestry polygenic risk score (PRS) of 269 variants and prostate cancer risk stratified by age. PRS associations in men of European ancestry (**A**) were meta-analyzed from UK Biobank (6852 cases and 193,117 controls) and Million Veteran Program (MVP) (13,643 cases and 210,214 controls; *Figure 2—figure supplement 1*), whereas PRS associations in men of African ancestry (**B**) were estimated from MVP (6353 cases and 53,362 controls). The x-axis indicates the PRS category. Additional analyses were performed to evaluate the PRS association in men with extremely high genetic risk (top percentile, 99–100%). The y-axis indicates the OR with error bars representing the 95% CIs for each PRS category compared to the 40–60% PRS category. The dotted horizontal line corresponds to an OR of 1. The number of cases and controls, ORs, and 95% CIs for each PRS category in each age stratum are provided in *Figure 2—source data 1*.

The online version of this article includes the following source data and figure supplement(s) for figure 2:

**Source data 1.** Association of multi-ancestry polygenic risk score (PRS) and prostate cancer risk stratified by age.

**Figure supplement 1.** Association between the multi-ancestry polygenic risk score (PRS) of 269 variants and prostate cancer risk stratified by age in men of European ancestry from UK Biobank (**A**) and Million Veteran Program (MVP) (**B**).

for younger men supports previous studies (*Conti et al., 2021*), suggesting that the contribution of genetic factors to prostate cancer is greater at younger ages and that age needs to be considered when comparing PRS findings across studies and populations.

The PRS is an effective risk stratification tool for prostate cancer at both ends of the risk spectrum. Current guidelines consider age, self-reported race, and a family history of prostate cancer in PSA screening decisions (*Schaeffer et al., 2021*). Although the PRS generally did not differentiate aggressive versus nonaggressive prostate cancer, a substantial fraction of men who will develop aggressive tumors (~40%) are among a subset of men in the population with the highest PRS (top 20%; *Appendix 1—table 2*), while only ~7% of men who will develop aggressive tumors are among the subset of men in the population with the lowest PRS (bottom 20%; *Appendix 1—table 2*), suggesting

**Table 1.** Age at which 5% absolute risk of prostate cancer is reached in men from European, African, and Hispanic populations. Absolute risks of prostate cancer were estimated using age- and population-specific Surveillance, Epidemiology, and End Results (SEER) incidence rates, CDC National Center for Health Statistics mortality rates, and polygenic risk score (PRS) associations from *Figure 2—source data 1* based on Million Veteran Program (MVP) and the UK Biobank.

| PRS category | European | African | Hispanic |
|---|---|---|---|
| [0–10] | >85 | 74 | >85 |
| (10–20%] | 81 | 70 | 83 |
| (20–30%] | 75 | 67 | 77 |
| (30–40%] | 72 | 66 | 71 |
| (40–60%] | 69 | 63 | 71 |
| (60–70%] | 66 | 61 | 68 |
| (70–80%] | 65 | 59 | 66 |
| (80–90%] | 62 | 58 | 65 |
| (90–100%] | 57 | 55 | 60 |
| (99–100%] | 52 | 51 | 53 |

that reduced screening among low PRS men may reduce the overdiagnosis of prostate cancer. Indeed, previous studies in men of European ancestry support that PRS-stratified screening could significantly reduce the overdiagnosis of prostate cancer by 33–42%, with the largest reduction observed in men with lower genetic risk (*Pashayan et al., 2015a*; *Callender et al., 2019*; *Pashayan et al., 2015b*). Risk-stratified screening studies are warranted in diverse populations to evaluate the clinical utility of this multi-ancestry PRS for early disease detection and when in a man's life genetic risk should be considered in the shared decision-making process of prostate cancer screening.

## Materials and methods
### Participants and genetic data

We replicated the association between the multi-ancestry PRS and prostate cancer risk in three independent case–control samples from the VA MVP, the MADCaP Network, and the NCI-MD, as described below. Previously, this multi-ancestry PRS was replicated by our group and others in the CA UG (1586 cases and 1047 controls of African ancestry), the UK Biobank (6852 cases and 193,117 controls of European ancestry; updates to the UK Biobank led to slightly different sample sizes in this study of 8483 cases and 193,744 controls of European ancestry), and the MGB (formerly known as the Partners Healthcare Biobank, 67 cases and 457 controls of African ancestry and 1554 cases and 10,918 controls of European ancestry). Results from these studies are described in detail elsewhere (*Conti et al., 2021*; *Plym et al., 2022*). To provide a comprehensive assessment of the PRS validation, we meta-analyzed all replication studies, which included a total of 22,049 cases and 414,249 controls of European ancestry (UK Biobank, MGB Biobank, and MVP) and 8794 cases and 55,657 controls of

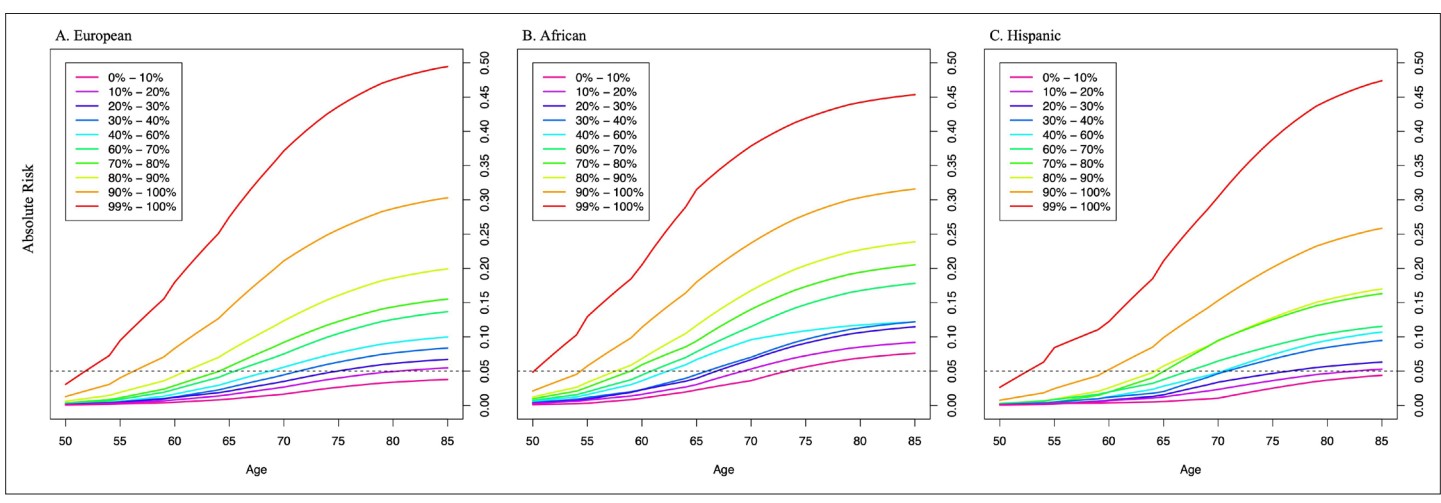

**Figure 3.** Absolute risk of prostate cancer by polygenic risk score (PRS) category in men from European (**A**), African (**B**), and Hispanic populations (**C**). The absolute risks were estimated using the age- and population-specific PRS associations from *Figure 2—source data 1*, the Surveillance, Epidemiology, and End Results (SEER) incidence rates, and the CDC mortality rates corresponding to non-Hispanic White, Black, and Hispanic men. The dotted line indicates the 5% absolute risk of prostate cancer.

African ancestry (MGB Biobank, MADCaP Network, NCI-MD, and MVP). In men of Hispanic ancestry, the multi-ancestry PRS was only assessed in MVP (1,082 cases and 20,601 controls).

All study protocols were approved by each site's Institutional Review Board, and informed consent was obtained from all study participants in accordance with the principles outlined in the Declaration of Helsinki.

## MVP

The design of the MVP has been previously described (*Gaziano et al., 2016*). Briefly, participants were recruited from approximately 60 Veteran Health Administration (VHA) facilities across the United States since 2011 with the current enrollment at >800,000. Informed consent was obtained for all participants to provide a blood sample for genetic analysis and access their full clinical and health data. The study received ethical and study protocol approval from the VA Central Institutional Review Board in accordance with the principles outlined in the Declaration of Helsinki.

A total of 485,856 samples from participants enrolled between 2011 and 2017 were genotyped on a custom Axiom array designed specifically for MVP (MVP 1.0). The genotyping array design and data quality controls were extensively described elsewhere (*Hunter-Zinck et al., 2020*). After excluding variants with high genotype missingness (>5%) and those that deviated from the expected allele frequency observed in the reference populations, genotype data were imputed to the 1000 Genomes Project Phase 3 reference panel (*1000 Genomes Project Consortium, 2015*). In MVP, genetic ancestry was assessed using HARE (*Fang et al., 2019*), which assigned >98% of participants with genotype data to one of four nonoverlapping population groups: non-Hispanic White (European), non-Hispanic Black (African), Hispanic, and non-Hispanic Asian. Due to the small number of non-Hispanic Asian individuals, they were excluded from the current analysis.

We identified a total of 21,078 cases and 284,177 controls from MVP, of whom 13,643 cases and 210,214 controls were of European ancestry (73.3%), 6353 cases and 53,362 controls were of African ancestry (19.6%), and 1082 cases and 20,601 controls were Hispanic (7.1%). Prostate cancer cases were identified from the Veterans Affairs Central Cancer Registry (VACCR), which collects cancer diagnosis, extent of disease and staging, first course of treatment, and outcomes from 132 VA medical centers. In this analysis, we only included cases from the VACCR who have a confirmed cancer diagnosis based on their diagnostic code, procedure code, and information from other clinical documents. Among the MVP participants without any prostate cancer diagnostic codes, we limited controls to those aged 45–95 years and had at least one prostate-specific antigen (PSA) test after enrollment. For prostate cancer cases, we obtained additional information on cancer staging and Gleason score to define aggressive prostate cancer phenotypes. Specifically, prostate cancer was considered aggressive if one of the following criteria was met: tumor stage T3/T4, regional lymph node involvement (N1), metastatic disease (M1), or Gleason score ≥8.0. Nonaggressive cases were defined as tumor stage T1/T2 and Gleason score <7.

## MADCaP

The MADCaP Network dataset included 405 prostate cancer cases and 396 controls from sub-Saharan Africa, as previously described (*Harlemon et al., 2020*; *Andrews et al., 2018*), with a substantial proportion of cases diagnosed at late stages. The study protocol was approved by each study site's Institutional Review Board/Ethnic Review Board. Written informed consent was obtained from all participants, and studies were conducted in concordance with the Declaration of Helsinki and the U.S. Common Rule. The MADCaP samples were genotyped on a customized array designed to capture common genetic variation in diverse African populations, and genotyping and quality control have been described in detail elsewhere (*Harlemon et al., 2020*). GWAS data were imputed using the 1000 Genomes Project Phase 3 reference panel (*1000 Genomes Project Consortium, 2015*).

## NCI-MD

The NCI-MD Study included 383 prostate cancer cases identified from two Maryland hospitals and 395 population-based controls from Maryland and its neighboring states (*Smith et al., 2017*). The study was approved by the NCI (protocol # 05C-N021) and the University of Maryland (protocol #0298229) Institutional Review Boards. Informed consent was obtained from all participants. About 87% of the cases in this study were considered nonaggressive, with pathologically confirmed T1 or

T2 tumor and a Gleason score ≤7. All samples from this study were genotyped on the Illumina InfiniumOmni5Exome array and were imputed to the 1000 Genomes Project Phase 3 reference panel (*1000 Genomes Project Consortium, 2015*).

## PRS construction and association analyses

PRSs were constructed by summing variant-specific weighted allelic dosages from 269 previously identified prostate cancer risk variants (*Conti et al., 2021*). Variants were weighted using the multi-ancestry conditional weights generated from our previous trans-ancestry GWAS for prostate cancer (*Conti et al., 2021*). Variants and weights used to generate the PRS can be found in the PGS Catalog: https://www.pgscatalog.org/publication/PGP000122/.

The association of PRS on prostate cancer risk (i.e., case–control status) was estimated separately in each replication study using an indicator variable for the percentile categories of the PRS distribution: [0–10%], [10%–20%], [20%–30%], (30%–40%], (40%–60%], (60%–70%], (70%–80%], (80%–90%], and (90%–100%], where parentheses indicate greater than and square brackets indicate less than or equal to. Additional analysis was performed to obtain the association for the top 1% PRS by splitting the top PRS decile into (90%–99%] and (99%–100%] categories. PRS thresholds were determined in the observed distribution among controls in each study. In all replication studies, logistic regression was performed with the case–control status as the outcome (a binary dependent variable) and the PRS categories as independent predictors, adjusting for age and the up to 10 principal components of ancestry, with the (40%–60%] category as the reference. Age was defined as age at diagnosis for prostate cancer cases and age at last PSA testing (MVP) or age at study recruitment (MADCaP and NCI-MD) for controls.

Discriminative ability was evaluated in MVP by estimating the AUC for logistic regression models of prostate cancer that included covariates only (age and four principal components of ancestry) and for models that additionally included the PRS. All analyses were performed separately within each population.

We performed a fixed-effects inverse-variance-weighted meta-analysis to combine the ORs and standard errors for each PRS decile from individual replication studies by ancestry using R package *meta* (*Schwarzer et al., 2015*). This meta-analysis was conducted across the three studies of European ancestry, UK Biobank, MGB Biobank, and MVP, as well as across the five studies of African ancestry, MGB Biobank, CA UG, MADCaP Network, NCI-MD, and MVP.

In the two large replication studies, UK Biobank and MVP, logistic regression analyses were repeated stratifying both cases and controls at ages ≤55, (55–60], (60–65], (65–70], and >70, with adjustments for age (as a continuous variable) and the top principal components of ancestry. The PRS associations estimated in men of European ancestry from UK Biobank and MVP were meta-analyzed using a fixed-effects inverse-variance-weighted method. Heterogeneity between studies and across strata was assessed via a Q statistic between effects estimates with corresponding tests of significance (*Schwarzer et al., 2015*).

In the three ancestry populations from MVP, we also performed stratified analyses by disease aggressiveness, where cases were stratified as aggressive or nonaggressive and all controls were used in the corresponding stratified analysis. In both the aggressive cases vs. controls and nonaggressive cases vs. controls analyses, logistic regression was performed with the case–control status as the outcome (a binary dependent variable) and the PRS categories as independent predictors, adjusting for age and the up to 10 principal components of ancestry, with the (40–60%] category as the reference. Heterogeneity across strata was assessed via a Q statistic between effects estimates with corresponding tests of significance (*Schwarzer et al., 2015*).

## Estimation of absolute risk

The absolute risk of prostate cancer was calculated for a given age for each PRS category in European, African, and Hispanic ancestry men (*Antoniou et al., 2010*; *Kuchenbaecker et al., 2017*; *Amin Al Olama et al., 2015*; *Antoniou et al., 2001*). The approach constrains the PRS-specific absolute risks for a given age to be equivalent to the age-specific incidences for the entire population, such that age-specific incidence rates are calculated to increase or decrease based on the estimated risk of the PRS category and the proportion of the population within the PRS category. The calculation accounts for competing causes of death.

Specifically, for a given population and PRS category $k$ (e.g., 80–90%, 90–100%), the absolute risk by age $t$ is computed as $AR_k(t) = \sum_0^t P_{ND}(t) S_k(t) I_k(t)$. This calculation consists of three components:

1. $P_{ND}(t)$ is the probability of not dying from another cause of death by age $t$ using age-specific mortality rates, $\mu_D(t)$ : $P_{ND}(t) = \exp\left[-\sum_0^t \mu_D(t-1)\right]$. In this analysis, the age-specific mortality rates from the National Center for Health Statistics, CDC (1999–2013) were used.
2. $S_k(t)$ is the probability of surviving prostate cancer by age $t$ in the PRS category $k$ and uses the prostate cancer incidence by age $t$ for category $k$: $S_k(t) = \exp\left[-\sum_0^t I_k(t-1)\right]$.
3. The prostate cancer incidence by age $t$ for PRS category $k$ is $I_k(t)$ and is calculated by multiplying the population prostate cancer incidence for the reference category, $I_0(t)$ and the corresponding risk ratio, $\beta_{ka}$, for PRS category $k$ and age category $a$ (e.g., ages ≤55, 55–60, 60–65, 65–70, and >70) containing age $t$. These are estimated from the odds ratio obtained from the population-specific individual-level PRS analysis for each age-stratum (African and Hispanic ancestry odds ratios from MVP and European ancestry odds ratios meta-analyzed from MVP and UK Biobank): $I_k(t) = I_0(t)\, exp\left(\beta_{ka}\right)$.

Prostate cancer incidence for age $t$ for the reference category, $I_0(t)$, was obtained by constraining the weighted average of the population cancer incidences for the PRS categories to the population age-specific prostate cancer incidence, $\mu(t)$. $I_0(t) = \mu(t) \frac{\sum_K f_k S_k(t-1)}{\sum_K f_k S_k(t-1) exp(\beta_k)}$, where $f_k$ is the frequency of the PRS category $k$ with $f_k = 0.1$ for all nonreference categories in our primary PRS analysis by deciles (e.g., 0–10%, 10–20%, 20–30%, etc.).

By leveraging the definition that $S_k(t=0) = 1$, for all $k$, the absolute risks were calculated iteratively by first getting $I_0(t=1)$, then $I_k(t=1)$, then $S_k(t=1)$, and finally $AR_k(t=1)$. Subsequent values were then calculated recursively for all $t$.

For each population, absolute risks by age $t$ were calculated using age- and population-specific prostate cancer incidence, $\mu(t)$, from the SEER program (1999–2013) and age- and population-specific mortality rates, $\mu_D(t)$, from the National Center for Health Statistics, CDC (1999–2013).

## Additional information

### Competing interests

Burcu F Darst: received honorarium for presentations at Society of Urology Oncology Annual Meeting (2021) and the Social Genomics Group at the University of Wisconsin, Madison (2021). The author has no other competing interests to declare. Ravi K Madduri: has stock or stock options in Navipoint Genomics LLC. The author has no other competing interests to declare. The other authors declare that no competing interests exist.

### Funding

| Funder | Grant reference number | Author |
|---|---|---|
| National Cancer Institute | U19 CA214253 | Christopher A Haiman |
| National Cancer Institute | U01 CA257328 | Christopher A Haiman |
| National Cancer Institute | U19 CA148537 | Christopher A Haiman |
| National Cancer Institute | R01 CA165862 | Christopher A Haiman |
| National Cancer Institute | K99 CA246063 | Burcu F Darst |
| National Cancer Institute | T32CA229110 | Fei Chen |
| Prostate Cancer Foundation | 20CHAS03 | Christopher A Haiman |
| Prostate Cancer Foundation | 21YOUN11 | Burcu F Darst |
| Achievement Rewards for College Scientists Foundation | | Burcu F Darst |

| Funder | Grant reference number | Author |
|---|---|---|
| Million Veteran Program | MVP017 | J Michael Gaziano
Amy C Justice |

The funders had no role in study design, data collection and interpretation, or the decision to submit the work for publication.

## Author contributions

Fei Chen, Burcu F Darst, Formal analysis, Investigation, Visualization, Writing – original draft, Writing – review and editing; Ravi K Madduri, Alex A Rodriguez, Xin Sheng, Data curation, Formal analysis, Project administration; Christopher T Rentsch, Wei Tang, Adam S Kibel, Anna Plym, Data curation, Writing – review and editing; Caroline Andrews, Data curation, Project administration; Kelly Cho, Mohamed Jalloh, Serigne Magueye Gueye, Lamine Niang, Olufemi J Ogunbiyi, Olufemi Popoola, Akindele O Adebiyi, Oseremen I Aisuodionoe-Shadrach, Hafees O Ajibola, Mustapha A Jamda, Olabode P Oluwole, Maxwell Nwegbu, Ben Adusei, Sunny Mante, Afua Darkwa-Abrahams, James E Mensah, Andrew Anthony Adjei, Halimatou Diop, Joseph Lachance, Timothy R Rebbeck, Stefan Ambs, Data curation; J Michael Gaziano, Data curation, Funding acquisition; Amy C Justice, Resources, Data curation, Funding acquisition, Writing – review and editing; David V Conti, Data curation, Funding acquisition, Methodology, Writing – review and editing; Christopher A Haiman, Conceptualization, Resources, Data curation, Supervision, Funding acquisition, Writing – review and editing

## Author ORCIDs

Fei Chen ⓘ http://orcid.org/0000-0002-1679-9932
Ravi K Madduri ⓘ http://orcid.org/0000-0003-2130-2887
Olufemi J Ogunbiyi ⓘ http://orcid.org/0000-0002-8748-2879
Afua Darkwa-Abrahams ⓘ http://orcid.org/0000-0003-0649-3996
Joseph Lachance ⓘ http://orcid.org/0000-0002-4650-3741
Christopher A Haiman ⓘ http://orcid.org/0000-0002-0097-9971

## Ethics

Human subjects: All study protocols were approved by each site's Institutional Review Board, and informed consent was obtained from all study participants in accordance with the principles outlined in the Declaration of Helsinki.

## Decision letter and Author response

Decision letter https://doi.org/10.7554/eLife.78304.sa1
Author response https://doi.org/10.7554/eLife.78304.sa2

# Additional files

## Supplementary files
• MDAR checklist

## Data availability

This investigation included published results from the following studies under DOI numbers 10.1038/s41588-020-00748-0 and 10.1093/jnci/djab058. For the MVP data, the final data sets underlying this study cannot be shared outside the VA, except as required under the Freedom of Information Act (FOIA), per VA policy. However, upon request through the formal mechanisms in place and pending approval from the VHA Office of Research Oversight (ORO), a de-identified, anonymized dataset underlying this study can be created. Upon request through the formal mechanisms provided by the VHA ORO, we would be able to provide sufficiently detailed variable names and definitions to allow replication of our work. Any requests for data access should be directed to the VHA ORO ( OROCROW@va.gov), and should reference the following project and analysis: 'MVP017: A VA-DOE Exemplar Project on Cancer'. Publicly available data described in this manuscript can be found from the following websites: 1000 Genomes Project (https://www.internationalgenome.org/); SEER (https://seer.cancer.gov/); National Center for Health Statistics, and CDC (https://www.cdc.gov/nchs/index.htm).

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

# Appendix 1

**Appendix 1—table 1.** Model discrimination and improvement estimated with area under the curve (AUC) upon adding the multi-ancestry polygenic risk score (PRS) to a base model in the Million Veteran Program (MVP) study populations.

| Population | Sample | Age and PCs | | Age, PCs, and PRS | | |
|---|---|---|---|---|---|---|
| | | AUC | 95% CI | AUC | 95% CI | AUC Change |
| European ancestry | All cases and controls | 0.582 | (0.578–0.587) | 0.694 | (0.690–0.699) | +0.112 |
| | Aggressive cases and controls | 0.533 | (0.521–0.545) | 0.666 | (0.655–0.677) | +0.133 |
| | Nonaggressive cases and controls | 0.603 | (0.598–0.608) | 0.703 | (0.698–0.708) | +0.100 |
| African ancestry | All cases and controls | 0.512 | (0.505–0.520) | 0.656 | (0.649–0.663) | +0.144 |
| | Aggressive cases and controls | 0.547 | (0.531–0.564) | 0.681 | (0.665–0.697) | +0.134 |
| | Nonaggressive cases and controls | 0.522 | (0.514–0.529) | 0.657 | (0.649–0.665) | +0.135 |
| Hispanic | All cases and controls | 0.530 | (0.513–0.547) | 0.683 | (0.667–0.699) | +0.153 |
| | Aggressive cases and controls | 0.568 | (0.531–0.607) | 0.674 | (0.636–0.712) | +0.106 |
| | Nonaggressive cases and controls | 0.514 | (0.495–0.534) | 0.685 | (0.667–0.702) | +0.171 |

Abbreviation: PCs, principal components of ancestry.

**Appendix 1—table 2.** The association between the multi-ancestry polygenic risk score (PRS) and prostate cancer aggressiveness in Million Veteran Program (MVP) participants from European, African, and Hispanic populations.
PRS categories were determined based on the distribution in controls in each replication study. ORs and 95% CIs were estimated from logistic regression models adjusting for age and principal components of ancestry. Heterogeneity was assessed via a Q statistic between effects estimates with corresponding tests of significance.

| PRS category | Aggressive cases vs. controls | | | | | Nonaggressive cases vs. controls | | | | | P-heterogeneity |
|---|---|---|---|---|---|---|---|---|---|---|---|
| | Controls | Cases | OR | 95% CI | p-Value | Controls | Cases | OR | 95% CI | p-Value | |
| European ancestry | | | | | | | | | | | |
| [0–10%] | 21,022 | 82 | 0.47 | (0.37–0.59) | 5.16E-10 | 21,022 | 258 | 0.31 | (0.27–0.36) | 4.43E-66 | 4.86E-03 |
| (10–20%] | 21,021 | 96 | 0.55 | (0.44–0.68) | 1.63E-07 | 21,021 | 423 | 0.51 | (0.46–0.57) | 6.29E-34 | 0.61 |
| (20–30%] | 21,021 | 118 | 0.67 | (0.54–0.83) | 1.76E-04 | 21,021 | 520 | 0.63 | (0.57–0.70) | 1.45E-19 | 0.60 |
| (30–40%] | 21,022 | 156 | 0.89 | (0.73–1.07) | 2.14E-01 | 21,022 | 656 | 0.79 | (0.72–0.87) | 8.87E-07 | 0.30 |
| (40–60%] | 42,042 | 352 | 1.00 (ref.) | | | 42,042 | 1658 | 1.00 (ref.) | | | |
| (60–70%] | 21,022 | 245 | 1.39 | (1.18–1.64) | 7.32E-05 | 21,022 | 1120 | 1.35 | (1.25–1.45) | 6.55E-14 | 0.71 |
| (70–80%] | 21,021 | 272 | 1.55 | (1.32–1.82) | 7.13E-08 | 21,021 | 1392 | 1.67 | (1.55–1.79) | 1.84E-42 | 0.41 |
| (80–90%] | 21,021 | 335 | 1.91 | (1.64–2.22) | 4.51E-17 | 21,021 | 1801 | 2.15 | (2.00–2.30) | 5.69E-105 | 0.16 |
| (90–100%] | 21,022 | 554 | 3.17 | (2.77–3.63) | 2.72E-63 | 21,022 | 3151 | 3.71 | (3.48–3.94) | <4.35E-283 | 0.04 |
| (99–100%] | 2,103 | 112 | 6.49 | (5.22–8.07) | 1.08E-63 | 2103 | 589 | 6.77 | (6.10–7.51) | 4.35E-283 | 0.73 |
| African ancestry | | | | | | | | | | | |
| [0–10%] | 5,337 | 29 | 0.35 | (0.24–0.53) | 2.96E-07 | 5337 | 163 | 0.35 | (0.30–0.42) | 2.50E-33 | 0.98 |

*Appendix 1—table 2 Continued on next page*

Appendix 1—table 2 Continued

| | Aggressive cases vs. controls | | | | | Nonaggressive cases vs. controls | | | | | |
|---|---|---|---|---|---|---|---|---|---|---|---|
| (10–20%] | 5,336 | 45 | 0.55 | (0.40–0.77) | 4.23E-04 | 5336 | 247 | 0.54 | (0.46–0.62) | 2.28E-17 | 0.88 |
| (20–30%] | 5,336 | 45 | 0.55 | (0.40–0.77) | 4.98E-04 | 5336 | 306 | 0.66 | (0.58–0.76) | 1.74E-09 | 0.33 |
| (30–40%] | 5,336 | 70 | 0.86 | (0.65–1.14) | 3.05E-01 | 5336 | 318 | 0.69 | (0.61–0.79) | 3.43E-08 | 0.16 |
| (40–60%] | 10,672 | 163 | 1.00 (ref.) | | | 10,672 | 920 | 1.00 (ref.) | | | |
| (60–70%] | 5,336 | 121 | 1.50 | (1.18–1.90) | 8.14E-04 | 5336 | 556 | 1.21 | (1.08–1.35) | 7.65E-04 | 0.11 |
| (70–80%] | 5,336 | 131 | 1.62 | (1.28–2.04) | 4.97E-05 | 5336 | 659 | 1.43 | (1.29–1.59) | 1.88E-11 | 0.36 |
| (80–90%] | 5,336 | 151 | 1.89 | (1.51–2.36) | 2.59E-08 | 5336 | 819 | 1.78 | (1.61–1.97) | 9.21E-30 | 0.64 |
| (90–100%] | 5,337 | 262 | 3.31 | (2.71–4.03) | 4.66E-32 | 5337 | 1,224 | 2.66 | (2.43–2.92) | 8.99E-97 | 0.05 |
| (99–100%] | 534 | 45 | 5.840 | (4.14–8.22) | 5.79E-24 | 534 | 220 | 4.77 | (4.02–5.66) | 1.62E-71 | 0.30 |
| **Hispanic** | | | | | | | | | | | |
| [0–10%] | 2061 | 2 | 0.12 | (0.03–0.50) | 3.69E-03 | 2061 | 21 | 0.31 | (0.20–0.50) | 8.67E-07 | 0.21 |
| [10–20%] | 2060 | 6 | 0.36 | (0.15–0.87) | 2.23E-02 | 2060 | 31 | 0.46 | (0.31–0.69) | 1.29E-04 | 0.61 |
| (20–30%] | 2060 | 6 | 0.36 | (0.15–0.86) | 2.12E-02 | 2060 | 47 | 0.70 | (0.50–0.98) | 4.00E-02 | 0.16 |
| (30–40%] | 2060 | 17 | 1.04 | (0.58–1.87) | 9.04E-01 | 2060 | 59 | 0.88 | (0.65–1.21) | 4.39E-01 | 0.64 |
| (40–60%] | 4,120 | 33 | 1.00 (ref.) | | | 4120 | 133 | 1.00 (ref.) | | | |
| (60–70%] | 2060 | 20 | 1.21 | (0.69–2.11) | 5.05E-01 | 2060 | 85 | 1.28 | (0.97–1.68) | 8.50E-02 | 0.87 |
| (70–80%] | 2060 | 24 | 1.47 | (0.86–2.49) | 1.55E-01 | 2060 | 136 | 2.06 | (1.61–2.63) | 7.82E-09 | 0.26 |
| (80–90%] | 2060 | 22 | 1.33 | (0.77–2.29) | 3.01E-01 | 2060 | 136 | 2.05 | (1.61–2.62) | 8.69E-09 | 0.15 |
| (90–100%] | 2060 | 31 | 1.92 | (1.17–3.15) | 9.37E-03 | 2060 | 217 | 3.30 | (2.64–4.12) | 7.46E-26 | 0.05 |
| (99–100%] | 206 | 4 | 2.580 | (0.91–7.38) | 7.61E-02 | 206 | 46 | 7.15 | (4.96–10.3) | 3.99E-26 | 0.07 |

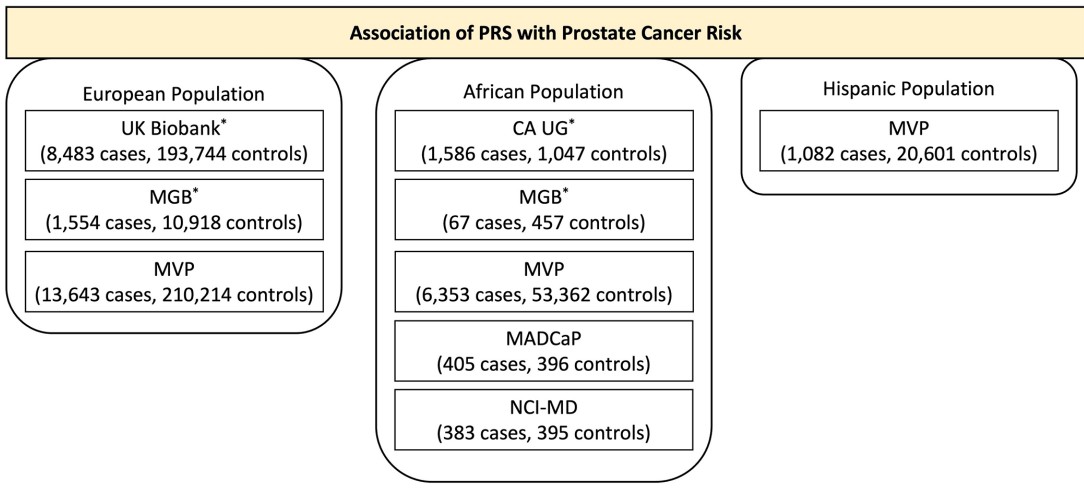

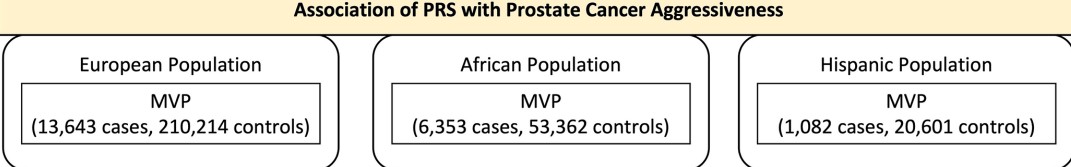

**Appendix 1—figure 1.** Individual studies of European, African, or Hispanic population included in the polygenic risk score (PRS) association analysis. Results from previous replication studies (*) in UK Biobank, Mass General Brigham (MGB) Biobank, and California and Uganda Prostate Cancer Study (CA UG) were meta-analyzed with results from Million Veteran Program (MVP), Maryland Prostate Cancer Case–Control Study (NCI-MD), and Men of African Descent and Carcinoma of the Prostate (MADCaP) Network within each ancestry population.

