## [Editor Report]

This article is mainly for an audience of genetic epidemiologists interested in the evaluation and portability of polygenic scores. The authors rigorously estimate the association of their multi-ancestry polygenic risk scores (PRS) for prostate cancer across multiple ancestries in a meta-analysis and show effect modification by age. The authors show that their PRS is effective in risk stratification for prostate cancer.

---

## [Decision Letter]

**Decision letter after peer review:**

Thank you for submitting your article "Validation of a Multi-Ancestry Polygenic Risk Score and Age-Specific Risks of Prostate Cancer: A Meta-analysis Within Diverse Populations" for consideration by *eLife*. Your article has been reviewed by 2 peer reviewers, and the evaluation has been overseen by a Reviewing Editor and Ricardo Azziz as the Senior Editor. The following individual involved in the review of your submission has agreed to reveal their identity: Arjun Bhattacharya (Reviewer #2).

Essential revisions:

The reviewers have agreed that the work presented here is an important contribution to the literature on polygenic risk scores and their applicability to cohorts of different ancestries. They also have agreed that the manuscript is clear, strong, and informative, and have only a small number of suggestions for improving clarity.

1. A reviewer mentions that added context for PRS is needed in the introduction. At the moment, the paper seems written for cancer epidemiology or genetic epidemiology audience, but it would be helpful if it were clear for a broader life sciences audience, who may not be completely aware of the utility of a PRS or may have preconceived notions about PRS research. Could the authors please add explanations and/or figures in the Introduction for:

– Explaining what a PRS is,

– what a PRS estimates,

– how the specific PRS was trained,

– the utility of a PRS in a clinical setting.

2. Can the authors also please briefly state how "prostate cancer risk" is calculated, either in the main text or in "Materials and methods" to guide the reader?

3. Could the authors please address how the results would be different if a random-effects meta-analysis model were to be used? A reviewer mentions that there is substantial heterogeneity in the environmental effects on prostate cancer risk across these different populations.

4. Have the authors looked into the PRS performance across ancestry proportion estimates and PRS associations with prostate cancer susceptibility across bins of ancestry proportions (i.e., AFR ancestry 0-20%, 20-40%, etc)? Could they comment a little on this topic? This could be an interesting follow-up.

5. Would it make sense to model select on other demographic/clinical covariates (e.g., SES measures) when estimating the odds ratios?

Other details

1. Can the authors please clarify what is the difference between Figure 1 and Figure 1 – source data 1? The caption of Figure 1 states that "ORs and 95% CIs for each PRS category are provided in Figure 1 – source data 1." but they seem to be already present in Figure 1. Also, please clarify the difference between Figure 2 and Figure 2 – source data 1.

2. The acronym "Partners" is used in the legend of Figure 1 instead of the acronym "MGB" used in the main text.

3. Page 20.- "Age-specific mortality rates are provided from a reference cohort." A reviewer recommends specifying which reference cohort is used.

---

## [Author Response]

Essential revisions:The reviewers have agreed that the work presented here is an important contribution to the literature on polygenic risk scores and their applicability to cohorts of different ancestries. They also have agreed that the manuscript is clear, strong, and informative, and have only a small number of suggestions for improving clarity.1. A reviewer mentions that added context for PRS is needed in the introduction. At the moment, the paper seems written for cancer epidemiology or genetic epidemiology audience, but it would be helpful if it were clear for a broader life sciences audience, who may not be completely aware of the utility of a PRS or may have preconceived notions about PRS research. Could the authors please add explanations and/or figures in the Introduction for:– Explaining what a PRS is,– what a PRS estimates,– how the specific PRS was trained,– the utility of a PRS in a clinical setting.

In response to this comment, in the Introduction section we include more background information regarding the development and validation of the multi-ancestry PRS for prostate cancer and what the PRS estimates (page 5). Further discussion on the clinical utility of this multi-ancestry PRS in identifying individuals at high risk of developing prostate cancer is included in the last paragraph of the Discussion section (page 9).

2. Can the authors also please briefly state how "prostate cancer risk" is calculated, either in the main text or in "Materials and methods" to guide the reader?

In our statistical analyses, “prostate cancer risk” refers to the relative risk estimated by the odds ratio calculated from a prostate cancer case-control analysis in each replication study. In all analyses assessing the association of PRS with prostate cancer risk, logistic regression models were used to estimate the odds ratio with case-control status as the outcome (a binary dependent variable) and the PRS categories as independent predictors, adjusting for age and up to ten principal components. The text in the Materials and methods section is modified to elaborate on these details in the statistical analysis (page 13 and page 14).

3. Could the authors please address how the results would be different if a random-effects meta-analysis model were to be used? A reviewer mentions that there is substantial heterogeneity in the environmental effects on prostate cancer risk across these different populations.

Within the European (UK Biobank, MVP, and MGB Biobank) and African population (MVP, MGB Biobank, MADCaP Network, CA UG, and NCI-MD) population, we also meta-analyzed the PRS associations with prostate cancer risk from individual replication studies using a random-effects method. We didn’t observe appreciable differences between the results from a fixed-effects and a random-effects meta-analysis in these two ancestry populations (see , Author response table 1). Given the similarities between these results, we only reported the PRS associations from the fixed-effects inverse-variance-weighted meta-analysis.

**Author response table 1. sa2table1:** Association of the Multi-ancestry PRS with Prostate Cancer Risk in men of European and African Ancestry. The PRS association on prostate cancer risk estimated from individual replication studies within each ancestry population were meta-analyzed using (a) a fixed-effects method and (b) a random-effects method.

	European Ancestry	African Ancestry
22,049 cases, 414,249 controls	8,794 cases, 55,657 controls
	Fixed-Effects Meta-analysis	Random-Effects Meta-analysis	Fixed-Effects Meta-analysis	Random-Effects Meta-analysis
PRS Category	OR	(95% CI)	P value	OR	(95% CI)	P value	OR	(95% CI)	P value	OR	(95% CI)	P value
[0–10%]	0.32	(0.29–0.35)	5.67E-126	0.32	(0.28–0.36)	3.72E-60	0.37	(0.32–0.43)	4.17E-41	0.39	(0.27–0.54)	6.49E-08
(10%–20%]	0.48	(0.44–0.51)	5.74E-77	0.47	(0.40–0.55)	2.49E-20	0.55	(0.48–0.62)	9.75E-22	0.58	(0.41–0.81)	1.58E-03
(20%–30%]	0.64	(0.59–0.68)	9.81E-35	0.64	(0.59–0.68)	6.29E-34	0.66	(0.59–0.74)	1.61E-12	0.65	(0.54–0.79)	6.85E-06
(30%–40%]	0.79	(0.75–0.85)	2.47E-12	0.78	(0.71–0.87)	1.12E-06	0.7	(0.62–0.78)	1.16E-10	0.7	(0.62–0.78)	8.81E-10
(40%–60%]	1.00 (reference)	1.00 (reference)	1.00 (reference)	1.00 (reference)
(60%–70%]	1.33	(1.26–1.40)	8.90E-24	1.32	(1.23–1.42)	5.44E-15	1.23	(1.11–1.35)	3.01E-05	1.26	(1.05–1.51)	1.09E-02
(70%–80%]	1.62	(1.54–1.71)	3.26E-72	1.61	(1.48–1.75)	5.21E-29	1.45	(1.32–1.59)	6.00E-16	1.47	(1.27–1.70)	3.13E-07
(80%–90%]	2.18	(2.08–2.29)	2.28E-216	2.16	(1.88–2.48)	4.28E-27	1.79	(1.64–1.95)	7.51E-40	1.8	(1.62–2.00)	3.36E-27
(90%–100%]	3.78	(3.62–3.96)	<5.00E-324	3.86	(3.52–4.23)	2.30E-184	2.8	(2.59–3.03)	1.38E-144	2.97	(2.57–3.44)	1.46E-49
(99%–100%]	7.32	(6.76–7.92)	<5.00E-324	7.7	(5.74–10.33)	3.28E-42	4.98	(4.27–5.79)	5.02E-95	5.17	(3.87–6.92)	1.57E-28

4. Have the authors looked into the PRS performance across ancestry proportion estimates and PRS associations with prostate cancer susceptibility across bins of ancestry proportions (i.e., AFR ancestry 0-20%, 20-40%, etc)? Could they comment a little on this topic? This could be an interesting follow-up.

We appreciate the reviewer’s insightful suggestion. In three of the five replication studies included in the African-ancestry meta-analysis (CA UG, MADCaP Network, and NCI-MD), we estimated the proportion of African ancestry (%AFR) from an unsupervised (K=2) ADMIXTURE analysis using the 1000 Genomes Project phase 3 European and African samples as the reference populations. We do not have access to the admixture information in the MVP and MGB Biobank data.

The %AFR was higher and less dispersed in MADCaP (mean = 95.7%, standard deviation [SD] = 4.6%) than in CA UG (mean = 79.2%, SD = 13.5%) and NCI-MD (mean = 77.2%, SD = 10.2%; see Author response table 2). Despite the differences in %AFR, we found that the PRS associations with prostate cancer risk were similar across these three studies, particularly in the top PRS decile (Figure 1 —figure supplementary 1). Although we were not able to formally test the effect modification of African ancestry on the PRS association due to the limited sample sizes of these three studies, the similar PRS associations observed in these studies support the robustness of this multi-ancestry PRS in risk stratification across African populations with varying degrees of admixture. Future investigations with sufficient sample sizes are warranted to better understand the interaction between admixture and PRS.

**Author response table 2. sa2table2:** 

	Sample Size	% AFR Ancestry
	No. Case	No. Control	Mean	SD
CA UG	1,586	1,047	79.2	13.5
MADCaP Network	405	396	95.7	4.6
NCI-MD	383	395	77.3	10.2

5. Would it make sense to model select on other demographic/clinical covariates (e.g., SES measures) when estimating the odds ratios?

For prostate cancer, only age, family history, and race are considered well-established risk factors. Given the potential correlation between family history and PRS, family history of prostate cancer is typically not adjusted for in the association analysis of PRS. Other demographic or clinical factors are unlikely to be associated with genotype status and thus not considered as potential confounders in the association between PRS and prostate cancer risk. Therefore, adjusting for these variables will not be expected to influence PRS associations.

Other details1. Can the authors please clarify what is the difference between Figure 1 and Figure 1 – source data 1? The caption of Figure 1 states that "ORs and 95% CIs for each PRS category are provided in Figure 1 – source data 1." but they seem to be already present in Figure 1. Also, please clarify the difference between Figure 2 and Figure 2 – source data 1.

Figure 1 is a graphic presentation of the results in Figure 1 – source data 1. Figure 1 – source data 1 provides the actual numbers of odds ratios (ORs), 95% confidence intervals (CIs), and P values for each PRS category in each ancestry population, which were cited in the main text. Although they present the same PRS association results, the detailed information included in Figure 1 – source data 1 would allow direct comparison or meta-analysis with other/future replication studies. The same applies to Figure 2 and Figure 2 – source data 2. We believe it is important to include both the figures and the tables (source data) in the manuscript to provide sufficient information for interpretation.

2. The acronym "Partners" is used in the legend of Figure 1 instead of the acronym "MGB" used in the main text.

We thank the reviewer for spotting this error. The legend of Figure 1 – source data 1 has been updated to be consistent with the main text, replacing “Partners” with “MGB Biobank” (page 22).

3. Page 20.- "Age-specific mortality rates are provided from a reference cohort." A reviewer recommends specifying which reference cohort is used.

We modified the text in the Materials and methods (page 15) to specify that the age-specific mortality rates from the National Cancer for Health Statistics, CDC (1993-2013) were used in the calculation of absolute risk of prostate cancer.